# Carbon Emission Factors Identification and Measurement Model Construction for Railway Construction Projects

**DOI:** 10.3390/ijerph191811379

**Published:** 2022-09-09

**Authors:** Xiaodong Hu, Ximing Zhang, Lei Dong, Hujun Li, Zheng He, Huihua Chen

**Affiliations:** 1School of Civil Engineering, Central South University, Changsha 410083, China; 2Longfor Group, Jianke Building, Xi’an 710000, China; 3School of Civil Engineering, Henan Polytechnic University, Jiaozuo 454003, China; 4School of Engineering Management, Hunan University of Finance and Economics, Changsha 410205, China

**Keywords:** railway construction projects, systematic literature review, semi-structured expert interview, carbon emission factors, carbon emission measurement model

## Abstract

Carbon emissions have become a focus of political and academic concern in the global community since the launch of the Kyoto Protocol. As the largest carbon emitter, China has committed to reaching the carbon peak by 2030 and carbon neutrality by 2060 in the 75th United Nations High-level Meeting. The transport sector needs to be deeply decarbonized in China to achieve this goal. Previous studies have shown that the carbon emissions of the railway sector are small compared to highways, waterways, and civil aviation. However, these studies only consider the operation stage and do not consider the carbon emissions caused by large-scale railway infrastructure construction during the construction stage. As an essential source of carbon emissions and the focus of emissions reduction, the carbon emission of railway construction projects (RCPs) is in urgent need of relevant research. Based on a systematic literature review (SLR), this paper sorts out the carbon emission factors (CEFs) related to RCPs; combines semi-structured expert interviews to clarify the carbon emissions measurement boundary of RCPs; modifies and calibrates CEFs; constructs the carbon emission measurement model of RCPs including building material production stage, building material transportation stage, and construction stage; and conducts empirical analysis to validate carbon emission factors and measurement models. This study effectively complements the theoretical research on CEFs and measurement models in the construction stage of railway engineering and contributes to guiding the construction of low-carbon railways practically.

## 1. Introduction

More than half of global greenhouse gas (GHG) emissions come from human activities and constitute a significant contributor to climate warming [1]. The transport sector accounted for around 24% of global carbon emissions from 2000 to 2018 and remains largely dependent on fossil fuels [2]. At present, as the country with the largest carbon emissions in the world, China’s emissions in 2018 reached 9.5 billion tons and accounted for about 28% of worldwide human emissions. It is a major area for carbon emissions reduction and low-carbon development. China aims to “strive to achieve carbon peak by 2030 and carbon neutrality by 2060” to combat climate change. The “14th Five-Year Plan” has also made comprehensive arrangements for achieving carbon peaking, carbon neutrality, and addressing climate change.

Carbon emissions from China’s transportation sector continue to grow rapidly. According to the calculation of energy type, the direct carbon emissions change of China’s transportation sector in 2018 was 980 million tons, of which road transportation, railway transportation, waterway transportation, and civil aviation transportation accounted for 73.5%, 6.1%, 8.9%, and 11.6%, respectively [3]. These data show that railway transportation accounts for the tiniest proportion. However, the above proportion only considers the operation stage and does not consider the carbon emissions caused by large-scale railway infrastructure construction during the construction stage. Railway construction consumes many building materials and is an essential source of carbon emissions [4]. Additionally, railways have strategic significance, and the scale of China’s railway industry continues to expand and grow steadily, which will inevitably lead to an increase in railway carbon emissions within a certain period. According to the 2020 statistical bulletin of China Railway Group, in 2020, the national railway fixed asset investment will be CNY 781.9 billion, and 4933 km of new lines will be put into operation, including 2521 km of high-speed railway (HSR). The national railway operating mileage is 146,300 km, including 38,000 km of HSR. The national railway network density is 152.3 km/10,000 square kilometers, the double-track rate is 59.5%, and the electrification rate is 72.8%. Therefore, the expansion of railway scale under the “carbon peak and carbon neutrality” goal must correctly handle the relationship between development and carbon reduction.

As a research hotspot, scholars have studied the carbon emissions of various transportation modes from different levels and proposed different carbon emission reduction measures and paths, which provide a good reference for this research. Currently, the research on carbon emission factors in engineering mainly involves energy, building materials, and construction equipment. Currently, widely used measurement methods include the life cycle assessment (LCA), input–output, field survey, carbon emission coefficient, material balance, and material flow analysis [5,6,7]. Scholars have proposed carbon emission reduction measures in the transportation sector from the national, provincial, and city levels, including carbon tax and the carbon emissions trading market [8]. However, the existing research lacks targeted research on RCPs, the boundary of railway carbon emission measurement is fuzzy, and there is a lack of systematic railway carbon emission factors. The availability of emission factor data is poor, and the measurement models differ significantly. Therefore, this paper clarifies the carbon emission measurement boundary, proposes a systematic carbon emission factor, constructs a carbon emission measurement model, and conducts a case analysis. This study can provide a reference for carbon emission reduction of railway projects and contribute to the realization of the dual carbon goal.

This paper is organized as follows. Section 2 sorts out the research status of railway carbon emissions, CEFs, and carbon emission measurement. Firstly, Section 3 identifies the CEFs of energy, building materials, and construction equipment based on SLR. Secondly, Section 4 clarifies the carbon emission measurement boundary of RCPs through semi-structured interviews and modifies and calibrates the CEFs of RCPs. Thirdly, Section 5 constructs the carbon emission measurement model of RCPs, including the building materials production stage, building materials transportation stage, and construction stage. Finally, Section 6 verifies the carbon emission factor and carbon emission measurement model proposed in this paper through the case of LN railway.

## 2. Literature Review

### 2.1. Railway Carbon Emission

Scholars have extensively researched the carbon emissions of different transportation modes in the past, but research on railway engineering is insufficient. Chester and Horvath [9] studied the life cycle energy consumption and carbon emissions of various transportation modes such as HSR, automobiles, heavy rail, and aircraft in the United States and analyzed the energy consumption factors and carbon emission factors per unit distance of different transportation modes and concluded that the energy consumption and greenhouse gas emissions of the HSR are the lowest. Chang and Kendall [10] combed the carbon emissions inventory of the infrastructure construction stage, and the results showed that nearly 80% of the carbon emissions were generated in the material production stage, and material transportation contributed 16% of the carbon emissions. Chang et al. [11] used a top-down modeling method to study China’s HSR’s energy and environmental footprint and counted the carbon emissions of roadbeds, tracks, bridges and culverts, tunnels, and four-electric material transportation and construction. The results show that the cradle-to-gate energy and GHG emissions of the HSR infrastructure are 104 and 9.2 million tons of CO_2_e, respectively, mainly from steel manufacturing. Lee et al. [12] quantified GHG emissions from construction modules covering earthworks, civil engineering activities (such as the construction of tunnels, viaducts, and bridges), railway tracks, passenger stations, and energy transmission and telecommunication systems for the entire HSR lines while identifying the key activity of each module and found about 92% of emissions originated from the use of materials. Pritchard and Preston [13] explored the contribution of tunnel engineering to the overall carbon emissions of railways based on multiple railway project cases and found that tunnels add significantly to both the embodied and operational energy consumption and GHG emissions of railway infrastructure. Kaewunruen et al. [14] summarized material characteristics and calculated the carbon emissions and energy consumption of material production for tunnel and rail construction. Additionally, the LCA analysis shows that the materials used in the construction process (i.e., tunnel and rail construction) contribute over 97% of the life cycle’s CO_2_ emissions [15].

### 2.2. Carbon Emission Factor

Existing scholars’ research on CEFs mainly includes energy, building materials, and construction equipment. Kim and Kim [16] proposed the life cycle EIA (LCEIA) method to identify changes in GHG emissions and presented the GHG emissions and ratios of different sources for sensitivity analysis. Ahmed et al. [17] estimated the carbon footprint of freight transport in Pakistan using a top-down energy-based method, proposing that the average carbon emission per liter of diesel combustion is 2.65 kg/L. Gao et al. [18] analyzed the influencing factors of cement carbon emissions. Wu et al. [19] analyzed and concluded that the carbonization of concrete and its disposal are the main links to generating carbon emissions. Pargana et al. [20] analyzed the environmental loads of various thermal insulation materials in the whole life cycle. Cao et al. [21] adopted the system dynamics model to simulate the carbon emission reduction scenario of the thermal coal supply chain’s four subsystems: production, transportation, storage, and consumption. Du et al. [22] listed the CEFs of different energy sources when studying the influences of different economic growth rates and policy factors of carbon emission reduction technologies on carbon emissions and carbon intensity of GDP by using the system dynamics method. Kaewunruen et al. [15] summarized 11 kinds of construction equipment for the track and the earthwork in the railway and calculated the energy consumption.

### 2.3. Carbon Emission Measurement

Due to the lack of literature on railway carbon emissions, this paper reviews the relatively mature literature on carbon emission measurement in the building industry. Currently, widely used carbon emission measurement methods include LCA, input–output method, carbon emission coefficient method, material balance method, and material flow analysis method. The carbon emission coefficient method and the input–output method are commonly used in the construction field.

Wu et al. [23] studied and analyzed the carbon footprint standards of the United States, Canada, Hong Kong, and other countries and regions, expounded on the idea of carbon footprint measurement, and concluded that the measurement boundary and evaluation indicators would affect the calculation results. Omar et al. [24] summarized the calculation methods of carbon emissions in the whole life cycle of materials and conducted empirical research based on data from different countries. Nassen et al. [25] compared the LCA method with the input–output analysis method and believed that the LCA method was suitable for the measurement of carbon emissions of individual buildings, while the input–output analysis method was more suitable for regional carbon emission research. Abanda et al. [26] studied the carbon emission measurement model of construction engineering. Moon et al. [27] conducted an empirical study, and after data comparison, they proposed that using the LCA method can significantly improve the accuracy of carbon emissions estimation in the design process. Wiedmann [28] reviewed and analyzed developed countries’ carbon emission research cases. Chen and Chen [29] constructed a multi-regional measurement model based on the input–output method and accumulated a large amount of research data. Tang et al. [30] focused on the implied energy consumption of imports and exports, established a multi-regional measurement model, and conducted a case study based in the UK.

With the rapid development of digital technology, Building Information Modeling (BIM) technology has gradually become an important auxiliary tool for carbon emission measurement in the whole life cycle of buildings. Regarding environmental impact assessments based on BIM, scholars have used BIM tools to study greenhouse gas emissions previously. El-Diraby et al. [31] fully realized the importance of the design scheme on the environmental impact of the whole life cycle and realized the coordination and communication between design and construction units in the design stage through collaborative tools. Inyim et al. [32] carried out a carbon emission estimation based on BIM. Isikdag and Underwood [33] creatively built a building construction information integration framework based on system-related theories. Singh et al. [34] focused on the information overlap and integration of various professional engineering projects in the United States and proposed the application of building information modeling technology to build a building information management platform.

### 2.4. Review of Existing Research

Carbon emissions-related research is mainly concentrated in the building industry. In the field of railway engineering, especially in the construction stage of RCPs, the related research is relatively scattered and not systematic. At present, there are still some deficiencies in the existing research: (1) The research boundary of carbon emission measurement of RCPs is ambiguous; (2) there is a lack of systematic CEFs of RCPs, and the availability of factor data is poor; and (3) the carbon emission measurement models vary greatly.

## 3. Identifying the CEFs Using SLR

### 3.1. SLR

SLR was employed in this study to identify the CEFs in prior studies, which can help researchers to reduce the omission of information and subjective errors. Two types of databases were selected to search the literature: Web of Science and CNKI. Initially, we chose three keywords, such as railway carbon emission, carbon emission factor, and carbon emission measurement for the search strategies, including the document title, abstract, and keywords from 2000 to 2022. The screening process began, including removing duplicate articles and filtering articles based on article titles, abstracts, and full-text relevance. Finally, we read the full text of the screened articles to sort out the relevant CEFs.

### 3.2. Energy CEFs

Energy CEFs mainly include fossil energy, electricity, biomass, hydrogen, nuclear energy, and other types. Bettle et al. [35] summarized the carbon emissions of different types of power plants in their study on the carbon emissions of the power system in England and Wales. Chen et al. [36] examined the power production mix of each state, approximated the carbon emission factor values of 50 states in the United States, and obtained carbon emission factors for coal, natural gas, biomass, and other energy sources. Chi et al. [37] based their findings on the building energy consumption data of China and 30 provinces from 2008 to 2018, where the carbon emission sources of the construction industry are defined as 11 primary energy sources (raw coal, briquette, coke, other coal preparation, other coking products, gasoline, kerosene, diesel, fuel oil, lubricating oil, and solvents oil). Ding et al. [38] based their findings on BIM and the carbon emission measurement model and listed common fuels and electricity carbon emission factors. The energy CEFs based on SLR are shown in Table 1.

### 3.3. Building Materials CEFs

Building materials CEFs mainly include aggregates, cement, concrete, steel, wood, and other materials. Crawford [45] evaluated the life cycle GHG emissions associated with wood and reinforced concrete railway sleepers and concluded that the life cycle emissions of reinforced concrete sleepers were six times less than those of wood sleepers. Lin et al. [41] used a hybrid input–output LCA method to calculate the carbon emissions from the different subsystems and sectors of the Beijing–Tianjin intercity HSR line, and the total emission caused by the construction stage was 3451.7 kt. Kaewunruen et al. [14] considered that carbon emissions and energy consumption mainly came from the production of building materials and the use of large-scale construction equipment during the construction stage, and calculated the percentage of carbon emissions of 13 materials in the construction stage of the entire rail system. Liljenstrom et al. [46] assessed the current annual climate impact and primary energy use of Swedish transport infrastructure using a methodological approach based on LCA and considered that steel and concrete accounted for about 75% of the climate impact of rail construction. The building materials CEFs based on SLR are shown in Table 2.

### 3.4. Construction Equipment CEFs

Construction equipment CEFs mainly include earth and rock-moving equipment, power equipment, hoisting equipment, transportation equipment, and other equipment types. Since the existing research on construction equipment CEFs is not systematic and not comprehensive enough, only some construction equipment CEFs can be listed. Kaewunruen et al. [15] summarized 11 kinds of construction equipment for the track and the earthwork in the railway and calculated the energy consumption. Ortega et al. [47] listed the values for the carbon analysis per kilogram/day per type of machine in the track. Lee et al. [12] present an estimation of GHG emissions at the construction stage of the infrastructure construction of the Honam HSR line from four stages of civil engineering works, railway tracks, stations, and energy transmission and telecommunication systems. Liu et al. [42] used a quota-based GHG emissions quantification model to calculate GHG emissions from all unit processes ahead of the operation of a subway station, which is classified into GHG emissions from construction material and those from construction equipment. The construction equipment CEFs based on SLR are shown in Table 3.

## 4. Modifying and Calibrating the Critical CEFs Using Semi-Structured Expert Interviews

### 4.1. Semi-Structured Interview Subjects

Based on the selection principles of respondents in previous studies [48,49], we went to five RCPs including Sichuan-Tibet Railway (CZ), Shanghai-Kunming Railway (HK), Kunming-Nanning Railway (YG), Huaihua-Shaoyang-Hengyang Railway (HSH), and Rizhao-Qufu Railway (LN) for research; covering areas including plains, mountains, and plateaus, and conducted semi-structured interviews with a total of 47 experts. These experts play an essential role in the RCPs and have clear positions, including 5 project managers, 4 chief engineers, 9 engineering (deputy) ministers, 10 safety and quality (deputy) ministers, and 19 other managers. In the semi-structured interview, the interviewer and the interviewee had an open-ended and face-to-face discussion, focusing on the current status and significant problems of carbon emission management in RCPs, measurement boundaries, critical CEFs, and management measures.

It should be noted that the CEFs screened based on SLR are not comprehensive enough, and the classification is not clear and systematic. Through semi-structured interviews, we added transportation CEFs to the CEFs for energy, building materials, and building equipment obtained through SLR. Further supplemented relevant literature such as statistical yearbooks, research reports, papers, and journal articles, revised the CEFs of RCPs, and completed the factor calibration, see Section 4.3 for details.

### 4.2. Measurement Boundary of Carbon Emissions of RCPs

According to the essential characteristics of railway engineering, combined with semi-structured expert interviews, this paper divides the carbon emission measurement boundary of RCPs into three stages, eight subsystems, and four types of CEFs. The three stages include building materials production, building materials transportation, and construction stage. The eight subsystems include bridges and culverts, tunnels, roadbeds, tracks, traction power supply, power, signal, and telecommunication (the latter four systems are generally referred to as four-electrical systems). Four types of CEFs include energy, transportation, building materials, and construction equipment, and the relationship between each factor is shown in Figure 1. However, because RCPs involve many specialties and a large amount of engineering, this paper only measures the carbon emissions of main energy, building materials, and construction equipment.

### 4.3. Modifying and Calibrating the CEFs

#### 4.3.1. CEFs Data Sources

There are three main sources of CEF data: data released by authoritative international organizations (such as 2019 Refinement to the 2006 IPCC Guidelines for National Greenhouse Gas Inventories), data released by relevant state departments (such as China Energy Statistical Yearbook), and research results of domestic scientific research institutions (such as Chinese Academy of Engineering, Carbon Emission Accounts & Datasets). This paper gives priority to domestic carbon emission data, and the priority of CEFs data sources is as follows: data released by relevant state departments > research results of domestic scientific research institutions > data released by authoritative international organizations.

#### 4.3.2. CEFs Calibration Method

Since it is not feasible to directly measure CEFs, scholars from relevant professional institutions have done much research on CEFs. Therefore, after reading a large amount of relevant literature on CEFs, the CEFs calibration in this paper will prioritize the research results of CEFs of authoritative institutions that conform to China’s national conditions and then make modifications based on the measurement boundary of this paper.

Because the CEFs measurement boundary in the references is not entirely consistent with this paper, and the CEFs may change over time. Therefore, it is necessary to modify the CEFs, clarify the measurement boundary of CEFs, and calibrate the CEFs according to the values in the related literature.

#### 4.3.3. Critical CEFs

Energy CEFs calibration

(1) Fossil energy CEFs

Fossil energy CEFs are calculated based on the calculation method of IPCC energy CEFs and China’s national conditions. Since the carbon emissions of fossil energy mainly come from the use (consumption) stage, and the carbon emissions in the production and transportation stage are difficult to measure, this paper only measures the carbon emissions generated in the use (consumption) stage.

According to the benchmark method published in the energy section of the 2019 Refinement to the 2006 IPCC Guidelines for National Greenhouse Gas Inventories, the formula for calculating the CEFs of fossil energy (combustion) in this paper is:(1)$$CEFs_{f}=V_{i}\times m_{i}\times cf\times u$$

${CEFs}_{f}$ is the carbon emission factors of fossil energy (kg CO_2_/unit); $V_{i}$ is the default net calorific value of fossil energy (kJ); $m_{i}$ is the default carbon content of fossil energy (kg C/GJ); cf is the default carbon oxide factor; u is the molar conversion factor of carbon and carbon dioxide. The calculation results of CEFs for critical fossil energy are shown in Table 4.

(2) Electricity CEFs

As clean energy, electricity does not directly generate carbon emissions during its use but consumes energy to generate carbon emissions during its production process. Therefore, the CEFs of electricity are affected by the energy structure of power generation. The Climate Change Department of the National Development and Reform Commission specified two types of marginal emission factors, OM (power marginal emission factor) and BM (capacity marginal emission factor) in the “2019 China Regional Power Grid Baseline Emission Factors” and unified the grid boundaries. It is divided into North China, Northeast China, Northwest China, East China, Central China, and Southern China regional power grids, excluding Tibet Autonomous Region, Taiwan Province, Hong Kong, and Macau Special Administrative Regions. The critical electricity CEFs in each region are shown in Table 5.

(3) Water CEFs

Water does not contain carbon, so water is not a direct carbon emission unit but indirectly generates carbon emissions by consuming energy during production and transportation. Therefore, the water CEFs refer to the carbon emission generated by the energy consumption per unit volume (mass) of water production and transportation. The value in the reference [53] is 0.91 kg CO_2_/m^3^. Since the density of water is 1000 kg/m^3^, the CEF of water can be converted to 0.00091 kg CO_2_/kg.

2.Transportation CEFs calibration

Combined with the actual situation of RCPs, this paper assumes that the transportation mode is road diesel transportation, and the CEF adopts the data of the literature [54], 2782.414 × 10^−4^ kg CO_2_/(t·km).

3.Building materials CEFs calibration

(1) Aggregate CEFs

Sand and gravel are essential building materials for railway construction, but there are few studies on carbon emissions during sand and gravel mining and processing. Therefore, this paper directly cites the CEFs in reference [54].

(2) Admixture CEFs

This paper directly cites the CEFs of bentonite, mineral powder, and fly ash from references [55,56].

(3) Cement CEFs

According to the literature [57], cement CEFs are the sum of raw material production stage CEFs, raw material transportation stage CEFs, and cement production and processing stage CEFs.

Add the CEFs of the raw material production stage, raw material transportation stage, and cement production and processing stage to obtain the total CEFs of cement. P.S.32.5, P.O. 42.5, and P.I.52.5 were 802.259 kg CO_2_/t, 1103.707 kg CO_2_/t and 1254.874 kg CO_2_/t respectively.

(4) Concrete CEFs

The calculation principle of the CEFs of concrete and cement is similar. Concrete CEFs are the sum of raw material production stage CEFs, raw material transportation stage CEFs, and concrete production and processing stage CEFs.

In the literature [58], the production amount and processing energy consumption of concrete with different strengths were obtained through field investigation, and the reliability was high. Therefore, this paper used seven strength grades of concrete in the reference (C20, C25, C30, C35, C40, C50, and C60) for raw material consumption and processing energy consumption.

The CEFs of concrete are obtained by adding up the CEFs of the raw material production stage, raw material transportation stage, and concrete processing production stage. The CEFs of the seven strength grades of concrete (C20–C60) are 306.192 kg CO_2_/m^3^, 336.680 kg CO_2_/m^3^, 371.654 kg CO_2_/m^3^, 389.568 kg CO_2_/m^3^, 419.599 kg CO_2_/m^3^, 502.819 kg CO_2_/m^3^ and 549.342 kg CO_2_/m^3^, respectively.

(5) Mortar CEFs

The calculation principle of the CEFs of mortar and cement is similar. Mortar CEFs are the sum of raw material production stage CEFs, raw material transportation stage CEFs, and mortar production and processing stage CEFs.

The CEFs of the mortar are obtained by adding the CEFs of the raw material production stage, the raw material transportation stage, and the mortar processing production stage. The CEFs of four different proportions of cement mortar are 829.388 kg CO_2_/m^3^, 636.038 kg CO_2_/m^3^, 581.347 kg CO_2_/m^3^, and 532.518 kg CO_2_/m^3^, respectively [53].

(6) Steel CEFs

Based on the classification of steels in [59], this paper appropriately expands the scope of steels. The CEFs of 4 types of steel are 3612.011 kg CO_2_/t, 2895.229 kg CO_2_/t, 3041.139 kg CO_2_/t and 3786.384 kg CO_2_/t, respectively.

(7) Wood CEFs

Since the measurement range of wood CEFs in the literature [60] is consistent with this paper, this paper cites the energy consumption of its wood production and processing process, as shown in Table 6, in which the energy CEFs adopt the value in this paper.

(8) Other building materials CEFs

Other building materials include waterproof coatings, modified bitumen waterproof membranes, waterproof boards, and rubber water stops. This paper cites references [59,61,62,63], and the calculated CEFs are 0.89 kg respectively CO_2_/kg, 3.53 kg CO_2_/m^2^, 19.553 kg CO_2_/m^2^ and 4.608 kg CO_2_/m.

The modified building materials CEFs in this paper are shown in Table 6.

4.Construction equipment CEFs calibrationn

The formula for calculating the CEFs of construction equipment is:(2)$$CEFs_{ci}=E_{i}\times CEFs_{ei}$$

$CEFs_{ci}\;$ is the carbon emission factors of construction equipment (kg CO_2_/per shift); $E_{i}$ is the energy consumption per shift of construction equipment (kg/kWh); and $CEFs_{ei}\;$ is the carbon emission factors of energy.

Among them, the construction equipment commonly used in RCPs comes from the “Basic Quota of Railway Engineering (2017)”; the energy consumption per shift of construction equipment comes from the “Railway Engineering Construction Equipment Shift Cost Quota (2017)”; and energy CEFs (gasoline, diesel, electricity, water, etc.) come from Section 4.3.3 in this paper. Table 7 shows the CEFs of commonly used construction equipment in RCPs.

## 5. Carbon Emission Measurement Model

### 5.1. Building Materials Production Stage

As shown in Figure 1, the building materials’ CEFs in this paper consider three stages of raw material acquisition, raw material transportation, and building materials production and processing. Therefore, the building materials’ CEFs can be directly used to calculate the carbon emissions in the building materials production stage.

Carbon emissions in building materials production stage = building materials consumption × (1 + building materials loss rate) × building materials production CEFs, summarizing all building materials calculation results to obtain building materials production stage carbon emissions, the calculation model is shown in Formula (3).
(3)$$C_{sc}=\sum_{i=1}^{n}m_{i}\times (1+u_{i})\times c_{i}$$

*C_sc_* is the carbon emission in the building materials production stage (kg CO_2_); *m_i_* is the consumption of building materials; *u_i_* is the loss rate of building materials (%); *c_i_* is the CEFs of building materials i in the production stage; and n is the number of building materials.

### 5.2. Building Materials Transportation Stage

The carbon emissions of building materials transportation refer to the carbon emissions generated by the consumption of gasoline, diesel, and other energy in the process of using vehicles to transport building materials from material production and processing sites to construction sites and transporting construction waste to landfills. In this paper, combined with the actual engineering situation, it is assumed that road diesel transportation is used for the transportation of building materials, and the transportation of building materials only needs to consider the material load, regardless of the material type.

Carbon emissions in the building materials transportation stage = building materials consumption (waste engineering volume) × (1 + loss rate) × transportation distance × transportation CEFs, and the carbon emissions in the building materials transportation stage are obtained by accumulating the transportation carbon emissions of various building materials (wastes), as shown in Formula (4).
(4)$$C_{ys}=\sum_{i=1}^{n}\sum_{j=1}^{k}m_{ij}\times (1+u_{i})\times d_{ij}c_{j}$$

*C_ys_* is the carbon emission during the transportation of building materials (kg CO_2_); *d_ij_* is the average transportation distance of building materials (wastes) i by transportation mode j; *m_i_* is the consumption of building materials (waste engineering volume); *u_i_* is the loss rate (%) of building materials (waste); *c_j_* is the transport CEFs per kilogram per kilometer of transport mode j; *n* is the number of types of building materials (wastes); and *k* is the number of modes of transportation.

### 5.3. Construction Stage

The carbon emissions in the construction stage consist of multiple parts: the carbon emissions generated by the energy consumption (gasoline, diesel, electricity, etc.) of the construction equipment, the carbon emissions generated by the amortization of the turnover materials, and the carbon emissions generated by the direct consumption of energy, as shown in the Formula (5).
(5)$$C_{js}=C_{js1}+C_{js2}$$

*C_js_* is the carbon emission in the construction stage; *C_js_*_1_ is the carbon emission of construction equipment (kg CO_2_); and *C_js_*_2_ is direct energy carbon emissions (kg CO_2_).

#### 5.3.1. Construction Equipment Carbon Emissions

In the construction stage, the consumption of equipment produces carbon emissions, while energy consumption by also equipment produces carbon emissions. Therefore, the carbon emissions of construction equipment can be calculated in two ways.

Construction equipment carbon emissions = Number of construction equipment shifts × Construction equipment CEFs = energy consumption × Energy CEFs

As shown in Formula (6).
(6)$$C_{js1}=\sum_{l=1}^{a}h_{l}\times c_{l}=\sum_{m=1}^{b}r_{m}\times c_{m}$$

*h_l_* is the number of shifts of construction equipment l; *c_l_* is the CEFs of construction equipment l unit shift; *r_m_* is the consumption of energy m; *c_m_* is the CEFs of energy m; *a* is the type of construction equipment; and *b* is the number of energy types.

The type of construction equipment and the number of shifts are taken from the quota, and the CEFs of construction equipment are from Table 7 in Section 4.3 of this paper.

#### 5.3.2. Direct Energy Carbon Emissions

In the construction stage, in addition to the indirect consumption of energy by equipment, there will also be direct energy consumption as shown in Formula (7).
(7)$$C_{js2}=\sum_{t=1}^{f}r_{t}\times c_{t}$$

*r_t_* is the consumption of energy m; *c_t_* is the CEFs of energy m; and *f* is the number of energy types.

The energy CEFs come from Section 4.3.3 in this paper.

## 6. Case Study

### 6.1. Project Overview and Data

LN railway is located in the southern part of Shandong Province, starting from Rizhao City in the east and passing through Linyi City to Qufu City, with a total design budget of CNY 33.691 billion. The length of the LN railway is 239.16 km, including 13.10 km of tunnels, 178.86 km of bridges, and 57.78 km of roadbeds. Due to the limited space of this article, only the main building materials data, building materials transportation data, mechanical shift data, and energy data that consume more are shown in Table 8, Table 9, Table 10 and Table 11.

### 6.2. The Value of Carbon Emissions

The total carbon emissions calculated from the two dimensions of the construction stage and the carbon emission source is 1,419,124.69 t. The total carbon emissions of materials is 1,253,677.42 t, of which the carbon emissions in the production stage is 1,124,700.91 t, and the carbon emissions in the transportation stage is 128,976.52 t. The total carbon emissions of construction equipment is 158,279.70 t. The total energy carbon emissions is 7167.56 t.

The material production stage is the largest source of carbon emissions in the case of railway construction, accounting for up to 79%; the carbon emissions in the construction and construction stage account for 9–12%. In the case of railway construction, material carbon emissions contributed 88%, construction equipment carbon emissions contributed 11%, and direct energy use only accounted for 1%.

## 7. Conclusions

This paper takes the carbon emissions of RCPs as the research object, calibrates the CEFs, builds a measurement model, and verifies the validity of the carbon emission factor and the model through case studies. The innovation of the research results lies in establishing systematic CEFs of RCPs, which are beneficial in promoting the carbon emission measurement of RCPs and low-carbon railway construction. The conclusions of this study are as follows: (1) Based on SLR, the energy, building materials, and construction equipment CEFs of RCPs are sorted out. (2) Combined with semi-structured expert interviews, the boundary of railway carbon emission measurement is clarified, the transportation CEFs are supplemented, and the calibration of energy, transportation, building materials, and equipment CEFs is completed. (3) The carbon emission measurement model of RCPs including the building materials production stage, building materials transportation stage, and construction stage is created. (4) The validity of the CEFs and measurement model is verified through a case study, and it is concluded that the building materials production stage is the largest carbon emission source in RCPs. This research is innovative and practical. This study effectively complements the theoretical research on CEFs and measurement models in the construction stage of RCPs, provides a basis for the measurement of carbon emissions and the analysis of emission reduction measures for RCPs, and contributes to guiding the construction of low-carbon railways from a practical level.

Since the use of materials accounts for the most significant proportion of the carbon emissions of RCPs, future research can consider the carbon emissions of new material inputs in the recycling stage. In addition, it is also possible to compare the differences in carbon emissions between different modes of transport and solve the problem of multimodal transport. Alternatively, at the regional or national level, we may analyze the carbon emissions of the entire life cycle of railway projects, explore the influencing factors of carbon emissions, and then propose more targeted carbon emission reduction paths.

## Figures and Tables

**Figure 1 ijerph-19-11379-f001:**
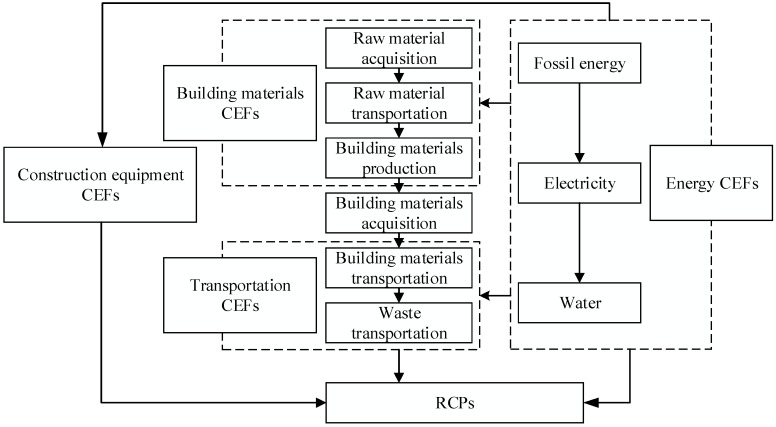
Relationship between energy, transportation, building materials, and construction equipment CEFs.

**Table 1 ijerph-19-11379-t001:** Energy CEFs based on SLR.

Energy Types	Source
Raw coal	Chi et al. [37]
Coal	Du et al. [22]; Chen et al. [36]; Bettle et al. [35]; Kiani et al. [39]
Other coal washings	Chi et al. [37]
Briquette	
Coke	Du et al. [22]; Chi et al. [37]; Kiani et al. [39]; Sarkara et al. [40]
Other coking products	Chi et al. [37]
Crude oil	Du et al. [22]
Gasoline	Du et al. [22]; Chi et al. [37]
Kerosene	
Diesel oil	Du et al. [22]; Chi et al. [37]; Ahmed et al. [17]; Lin et al. [41]
Solvent oil	Chi et al. [37]
Petroleum	Chen et al. [36]
Liquefied petroleum gas	Du et al. [22]
Fuel oil	Chi et al. [37]; Kiani et al. [39]
Lubricating oil	Chi et al. [37]
Natural gas	Du et al. [22]; Chen et al. [36]; Kiani et al. [39]
Electricity	Lin et al. [41]; Liu et al. [42]; Kiani et al. [39]; Liu et al. [43]
Biomass	Chen et al. [36]
Solar PV	Chen et al. [36]; Pu et al. [44]
Hydropower	Chen et al. [36]
Wind	
nuclear	

**Table 2 ijerph-19-11379-t002:** Building materials CEFs based on SLR.

Material Types	Source
Water	Chang et al. [11]; Kaewunruen et al. [14]; Lin et al. [41]
Sand	
Polyvinyl chloride (PVC) pipe	Chang et al. [11]; Lee et al. [12]
Polyethylene (PE) pipe	Chang et al. [11]; Kaewunruen et al. [14]
Waterproof rubber belt	Kaewunruen et al. [14]
Steel	Chang et al. [11]; Crawford [45]; Kaewunruen et al. [14]; Lee et al. [12]; Lin et al. [41]; Liu et al. [42]; Kim and Kim [16]; Liljenstrom et al. [46]
Geogrid	Chang et al. [11]
Rubble and gravel	Chang et al. [11]; Kaewunruen et al. [14]; Lin et al. [41]
Cement	Chang et al. [11]; Kaewunruen et al. [14]; Lee et al. [12]; Lin et al. [41]; Kim and Kim. [16]; Liljenstrom et al. [46]
Concrete	Kaewunruen et al. [14]; Lee et al. [12]; Liu et al. [42]; Kim and Kim. [16]; Liljenstrom et al. [46]
Accelerator	Kaewunruen et al. [14]
Binder	
Clay	Chang et al. [11]
Wood	Chang et al. [11]; Lin et al. [41]; Liu et al. [42]
Timber-hardwood	Crawford [45]
Lime	Chang et al. [11]
Bricks	
Cable	Lee et al. [12]; Lin et al. [41]

**Table 3 ijerph-19-11379-t003:** Construction equipment CEFs based on SLR.

Construction Equipment Types	Source
Concrete distributor	Kaewunruen et al. [15]
Concrete mixing plant	
CNC grinding machine	
Gantry crane	
Two-way transporter	
CA mortar truck	
Track laying machine	
Spiral drilling machine	
Excavator	
Loading machine	
Concrete pump	
Sleeper-laying machine	Ortega et al. [47]
Rail-laying machine	
Ballast-spreading machine	
Tamping machine	
Ballast-changing machine	
Ballast-cleaning machine	
Dump truck	Lin et al. [41]
Loader	
Crane	
Drill	
Bulldozer	
Water tank	
Air compressor	
Lift truck	
Concrete pump	
Generator	
Tractor	
Trainer	
Excavator	
Rammer	
Concrete transportation vehicle	Liu et al. [42]
Gantry crane	
Low-speed winch	
Steel binding machine	
Steel cutting machine	
Steel straightening machine	

**Table 4 ijerph-19-11379-t004:** Critical fossil energy CEFs.

Material Name	Default Net Calorific Value (kJ) [50]	Default Carbon Content (kg C/GJ) [51]	Default CO Factor	Unit	CEFs (kg CO_2_/Unit)
Raw coal	20,908	25.8	100%	kg	1.978
Clean coal	26,344	25.8	100%	kg	2.492
Coke	28,435	29.2	100%	kg	3.044
Crude	41,816	20.0	100%	kg	3.067
Kerosene	43,070	19.5	100%	kg	3.080
Gasoline	43,070	18.9	100%	kg	2.985
Diesel fuel	42,652	20.2	100%	kg	3.159
Liquefied Petroleum Gas	50,179	17.2	100%	kg	3.165
Natural gas	38,931	15.3	100%	m^3^	1.996
Coke oven gas	16,726	12.1	100%	m^3^	0.770

**Table 5 ijerph-19-11379-t005:** Critical electricity CEFs in six major regions in China in 2019.

Area	OM Electricity Carbon Emission Factor (kg CO_2_/kWh) [52]	BM Electricity Carbon Emission Factor (kg CO_2_/kWh) [52]
North China Power Grid	0.9419	0.4819
Northeast Power Grid	1.0826	0.2399
Northwest Power Grid	0.8922	0.4407
East China Power Grid	0.7921	0.3870
Central China Power Grid	0.8587	0.2854
China Southern Power Grid	0.8042	0.2135

**Table 6 ijerph-19-11379-t006:** Critical building materials CEFs.

Classification	Building Materials Name	Unit	CEFs (kg CO_2_/Unit)
Non-revolving material	Stone	m^3^	31.2
	Sand	m^3^	72.5
	Bentonite	kg	0.041
	Mineral powder	kg	0.05692
	Fly ash	kg	0.0015
	Ordinary Cement Grade 32.5	t	802.259
	Ordinary Cement Grade 42.5	t	1103.707
	Ordinary Cement Grade 52.5	t	1254.874
	C20 Concrete	m^3^	306.192
	C25 Concrete	m^3^	336.680
	C30 Concrete	m^3^	371.654
	C35 Concrete	m^3^	389.568
	C40 Concrete	m^3^	419.599
	C50 Concrete	m^3^	502.819
	C60 Concrete	m^3^	549.342
	1:1 Cement mortar	m^3^	829.388
	1:2 Cement mortar	m^3^	636.038
	1:2.5 Cement mortar	m^3^	581.347
	1:3 Cement mortar	m^3^	532.518
	Large steel	t	3612.011
	Middle and small steel	t	2895.229
	Hot rolled steel bar	t	3041.139
	Cold rolled steel bar	t	3786.384
	Iron product	t	2084.565
	Wood	m^3^	120.924
	Waterproof coating	kg	0.89
	Modified Asphalt Waterproof materials	m^2^	3.53
	PVC waterproof board	m^2^	19.553
	Rubber Waterstop	m	4.608
Reusable materials	Turnover Steel	t	1331.805
	Turnover wood	m^3^	12.092

**Table 7 ijerph-19-11379-t007:** Critical construction equipment CEFs.

Equipment Category	Construction Equipment	Energy Consumption per Shift			CEFs (kg CO_2_/per Shift)
		Gasoline (kg)	Diesel Fuel (kg)	Electricity (kWh)	
Earth-moving machinery	Crawler hydraulic single bucket excavator ≤ 0.4 m^3^	-	35.48	-	112.08
	Crawler hydraulic single bucket excavator ≤ 0.6 m^3^	-	44.08	-	139.25
	Crawler hydraulic single bucket excavator ≤ 1 m^3^	-	62.90	-	198.70
	Crawler Bulldozer ≤ 75 kW	-	49.73	-	157.10
	Crawler Bulldozer ≤ 300 kW	-	197.57	-	624.12
	Self-propelled vibratory roller ≤ 12 t	-	75.00	-	236.93
	Frog rammer ≤ 700 Nm	-	-	20.40	16.41
	Tire loader ≤ 2 m^3^	-	56.45	-	178.33
Hoisting Machinery	Crane ≤ 8 t	-	35.28	-	111.45
	Crane ≤ 16 t	-	57.15	-	180.54
	Gantry crane ≤ 10 t-22 m	-	-	61.44	49.41
	Gantry crane ≤ 20 t-22 m	-	-	109.44	88.01
	Gantry crane ≤ 50 t-40 m	-	-	176.64	142.05
	Crawler crane ≤ 10 t	-	31.75	-	100.30
	Crawler crane ≤ 15 t	-	38.81	-	122.60
	Crawler crane ≤ 40 t	-	47.63	-	150.46
	Crawler crane ≤ 250 t	-	352.80	-	1114.50
	Single drum slow speed winch ≤ 30 kN	-	-	38.40	30.88
	Single drum slow speed winch ≤ 50 kN	-	-	56.32	45.29
Transport machinery	Dump Truck ≤ 4 t	-	34.27	-	108.26
	Dump Truck ≤ 8 t	-	47.58	-	150.31
	Dump Truck ≤ 12 t	-	61.29	-	193.62
	Truck ≤ 4 t	26.61	-	-	79.43
	Truck ≤ 6 t	34.56	-	-	103.16
	Sprinkler ≤ 5000 L	34.56	-	-	103.16
	Small transporter ≤ 1 t	-	7.26	-	22.93
	Belt conveyor ≤ 10 m	-	-	15.36	12.35
	Concrete mixer truck ≤ 6 m^3^	-	88.70	-	280.20
	Concrete mixer truck ≤ 8 m^3^	-	100.80	-	318.43
	Concrete mixer truck ≤ 10 m^3^	-	106.85	-	337.54
Concrete and Mortar Machinery	Concrete mixer ≤ 250 L	-	-	15.68	12.61
	Concrete mixer ≤ 400 L	-	-	21.56	17.34
	Concrete mixer ≤ 800 L	-	-	86.24	69.35
	Concrete batching plant ≤ 60 m^3^/h	-	-	636.16	511.60
	Concrete batching plant ≤ 100 m^3^/h	-	-	913.92	734.97
	Concrete batching plant ≤ 120 m^3^/h	-	-	1008.00	810.63
	Concrete insertion vibrator	-	-	5.38	4.33
	Concrete attached vibrator	-	-	6.72	5.40
	Suspended pulp leveling machine	-	-	105.28	84.67
	Concrete wet spraying machine ≤ 5 m^3^/h	-	-	24.64	19.82
	Hydraulic grouting pump ≤ 50 L/min	-	-	30.80	24.77
	Concrete pump ≤ 60 m^3^/h	-	-	492.80	396.31
	Concrete delivery pump truck ≤ 90 m^3^/h	-	61.74	-	195.04
	Concrete placing machine ≤ 21 m	-	-	21.84	17.56
	Mortar mixer ≤ 200 L	-	-	13.44	10.81
	Mortar mixer ≤ 400 L	-	-	20.16	16.21
	Prestressed rebar hydraulic tensioning equipment ≤ 1200 kN	-	-	38.40	30.88
Foundation and Pump Machinery	Hydraulic vibratory pile driver ≤ 320 t	-	650.92	-	2056.26
	Hydraulic static pile driver ≤ 1200 kN	-	-	138.60	111.46
	Hydraulic static pile driver ≤ 1600 kN	-	-	189.00	151.99
	Impact hole forming machine d ≤ 1.5 m	-	-	177.60	142.83
	Crawler hydraulic grab grooving machine ≤ 1.2 m	-	194.04	-	612.97
	Dynamic compaction machinery ≤ 1200 kNm	-	56.70	-	179.12
	Dynamic compaction machinery ≤ 2000 kNm	-	69.30	-	218.92
	Single stage centrifugal water pump ≤ 12.5 m^3^/h-20 m	-	-	8.98	7.22
	Single stage centrifugal water pump ≤ 50 m^3^/h-38 m	-	-	44.88	36.09
	Multistage centrifugal clean water pump ≤ 85 m^3^/h-180 m	-	-	224.40	180.46
	Multistage centrifugal clean water pump ≤ 155 m^3^/h-185 m	-	-	538.56	433.11
	Sewage pump ≤ 90 m^3^/h-26 m	-	-	89.76	72.18
	Centrifugal mud pump ≤ 108 m^3^/h-21 m	-	-	89.76	72.18
	Slurry treatment centrifuge ≤ 100 m^3^/h	-	-	326.40	262.49
	Slurry separation equipment ≤ 1500 m^3^/h	-	-	1836.00	1476.51
	Slurry production cycle equipment ≤ 500 m^3^/h	-	-	136.00	109.37
welding machinery	AC arc welding machine ≤ 42 kVA	-	-	144.00	115.80
	DC arc welding machine ≤ 32 kW	-		102.40	82.35
	Butt welding machine ≤ 100 kVA	-	-	288.00	231.61
Paving machinery	Track laying machine 25 m	-	193.64	-	611.71
	Ballastless track long rail laying unit 500 m	-	94.25	-	297.74
	Turnout tamping car	-	298.17	-	941.92
	Long rail line laying and rolling mill	-	182.50	-	576.52
	Erecting machine ≤ 900 t	-	642.03	-	2028.17
	Box girder truck ≤ 900 t	-	913.92	-	2887.07
	Wheel-rail beam moving machine ≤ 900 t	-	188.50	-	595.47
	Wheel-rail beam lifter ≤ 2 × 450 t	-	188.50	-	595.47
	Track slab steel bar tensioning equipment	-	-	105.60	84.92
	Type I double block sleeper concrete pouring production line	-	-	384.00	308.81
Processing and other machinery	Steel bar straightening machine d ≤ 14	-	-	18.70	15.04
	Rebar cutting machine d ≤ 40	-	-	33.32	26.80
	Steel bending machine d ≤ 40	-	-	14.28	11.48
	Woodworking circular saw d ≤ 500	-	-	16.32	13.12
	Jaw Crusher ≤ 250 × 400	-	-	81.60	65.62
	Vertical drilling machine d ≤ 25	-	-	8.98	7.22
	Pipe cutting machine d ≤ 150	-	-	13.60	10.94

**Table 8 ijerph-19-11379-t008:** Main building materials data.

Serial Number	Material Name	Unit	Consumption
1	Ordinary cement grade 32.5	kg	123,992,044
2	Ordinary cement grade 42.5	kg	213,179,751
3	white cement	kg	13,420.27
4	double-fast cement	kg	1497.61
5	Ordinary Cement grade 42.5 (High Performance Concrete)	kg	248,951,256
6	log	m^3^	2151.04
7	sawn timber	m^3^	11,093.86
8	Water-emulsion polymer cement-based composite waterproof coating	kg	148.26
9	Water-based primer (common)	kg	176.04
10	Acrylic primer	kg	1287.21
11	Acrylic Elastic Premium Coating	kg	12,521.26
12	bentonite	kg	2,702,413.99
13	Slag Powder (High Performance Concrete)	kg	74,204,928.9
14	boulder	m^3^	10,368.25
15	flakes	m^3^	224,682.25

**Table 9 ijerph-19-11379-t009:** Building materials transportation data.

Serial Number	Material Name	Transport Distance (km)	Transport Mode
1	sand	80	Road Diesel Transport
2	stone	60.48	Road Diesel Transport
3	cement	203.38	Road Diesel Transport
4	steel	206.5	Road Diesel Transport
5	other metal materials	200	Road Diesel Transport
6	wood	65.74	Road Diesel Transport
7	other building materials	234.85	Road Diesel Transport
8	waste	1	Road Diesel Transport

**Table 10 ijerph-19-11379-t010:** Mechanical shift data.

Serial Number	Construction Equipment	Unit	Quantity
1	Crawler hydraulic single bucket excavator ≤ 0.6 m^3^	per shift	2090
2	Crawler hydraulic single bucket excavator ≤ 0.8 m^3^	per shift	17.56
3	Crawler hydraulic single bucket excavator ≤ 1.0 m^3^	per shift	539.11
4	Crawler hydraulic single bucket excavator ≤ 2.0 m^3^	per shift	3597.28
5	Crawler hydraulic single bucket excavator ≤ 2.5 m^3^	per shift	3.14
6	Crawler Bulldozer ≤ 75 kW	per shift	2737.35
7	Crawler Bulldozer ≤ 90 kW	per shift	430.11
8	Crawler Bulldozer ≤ 105 kW	per shift	2615.76
9	Static Roller ≤ 8 t	per shift	104.03
10	Static Roller ≤ 12 t	per shift	928.16
11	Static Roller ≤ 15 t	per shift	107.38
12	Self-propelled vibratory roller ≤ 0.7 t	per shift	364.58
13	Self-propelled vibratory roller ≤ 15 t	per shift	5440.65
14	Self-propelled vibratory roller ≤ 20 t	per shift	427.96
15	Grader ≤ 120 kW	per shift	790.86
16	frog rammer ≤ 250 Nm	per shift	113.56
17	frog rammer ≤ 700 Nm	per shift	3588.54
18	Stabilized soil mixer ≤ 120 kW	per shift	0.23
19	tire loader ≤ 2 m^3^	per shift	29,912.48
20	crawler loader ≤ 2 m^3^	per shift	4955.27

**Table 11 ijerph-19-11379-t011:** Energy data.

Serial Number	Energy Name	Unit	Quantity
1	coal tar	kg	11,988.08
2	coke	kg	386.04
3	coal	t	5.13
4	gasoline	kg	2013.35
5	kerosene	kg	239.7
6	water	t	2,064,227
7	electricity	kWh	6,508,482

## Data Availability

Not applicable.

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
