# Peer review of "Carbon Emission Factors Identification and Measurement Model Construction for Railway Construction Projects"

_ijerph, 2022, doi:10.3390/ijerph191811379_

Round 1

Reviewer 1 Report

-The introduction can be improved, arranged so that it flows and can be understood by the reader.

-Original and important research to the problem of climate change.

-The description of the method can be improved for better understanding.

Reviewer 2 Report

This study is to identify the influence factors of carbon emissions as well as construct the measurement model for railway construction projects, which is interesting and impressive. Some comments are as follows:

1. The authors are encouraged to perform empirical analysis or case study using the present calibration or simulation results, which is positive to address the importance and significance of this study.

2. The authors are recommended to put forward the implications of this study.

3. A comprehensive proofreading work is needed to improve written English in this manuscript.

Reviewer 3 Report

Remarks:

1) An interesting and substantive introduction to the manuscript. The research objectives of the study and research hypotheses were not given directly. I recommend that the authors complete this for better understanding by potential readers.

2) Literature review:

please describe the findings of the Chester and Horvath studies for the USA (lines: 81-86), Chang et al. (lines 90-92), Lee et al. (lines: 93-97), Chen et al. (faith: 97-98)

3) Explain the abbreviations - CZ, HK, YG, HSH and LN (line 225)

4) Expand the final conclusions, in particular regarding rail transport.

5) A weakness of the manuscript is the lack of empirical research (even examples of the use of formulas) and comparative analysis for different modes of transport.

Reviewer 4 Report

Dear authors

This paper presents the method of the quantitatively assessment for railway construction to considered carbon emission factors identification and measurement model construction.

We could understand the existing method because many existing research and literature were surveyed.

However, there are some points which are needed to confirm and correct before publication.

Those are:

1)    L61-L62 “As research hotspot …measures and paths”, I couldn’t understand the meaning. I think the authors must add the explanation about this part.

2)     In “2.1 Railway carbon emission” section, we could get the information the materials used stage in the construction process contribute over 97% of the total carbon emission (L101-102). Therefore, I think it is important to consider the recycle stage for new materials input. Is there any information in existing research or literature?

3)    In “2.3 Carbon emission measurement” section, “BIM” is used. I think this word is abbreviation of the “Building Information Modeling”. However, “BIM model” is written at L152. Is it correct word?

4)    At Table 1Table 3, it is difficult to confirm the each reference. If possible, please draw the line or change the style of the Table.

5)    In “4.3.3 Critical CEFs” section, the some of the calculation methods are confused (ex L287-288, “Fossil energy CEFs = default net calorific value × default carbon content × default carbon oxide factor × molar conversion factor of carbon and carbon dioxide”)

Please use the formula as in the 5.1 (1)

6)    Total of this paper, what is the novelty of this method? If authors developed new method, the quantitative results are needed. Is it possible to explain the difference of the existing method? If possible, please show the results with case study.  

I would like to confirm above the review and revise the manuscript.

Best regards,

Round 2

Reviewer 2 Report

The authors have addressed most of the comments.

Reviewer 4 Report

Dear authors

Thank you for sending the revise version.

I confirmed all comment was revised and modified precisely.

I felt the quality of the paper was much better than before one.

I hope this paper is published as the final version.

Best regards,